# The Effectiveness of Interactive Dashboards to Optimise Antibiotic Prescribing in Primary Care: A Systematic Review

**DOI:** 10.3390/antibiotics12010136

**Published:** 2023-01-10

**Authors:** Nathaly Garzón-Orjuela, Sana Parveen, Doaa Amin, Heike Vornhagen, Catherine Blake, Akke Vellinga

**Affiliations:** 1School of Public Health, Physiotherapy and Sports Science, University College Dublin, D04 V1W8 Dublin, Ireland; 2Insight Centre for Data Analytics, University of Galway, H91 AEX4 Galway, Ireland

**Keywords:** antibiotic prescribing, primary health care, general practice, clinical audit, dashboard

## Abstract

Governments and healthcare organisations collect data on antibiotic prescribing (AP) for surveillance. This data can support tools for visualisations and feedback to GPs using dashboards that may prompt a change in prescribing behaviour. The objective of this systematic review was to assess the effectiveness of interactive dashboards to optimise AP in primary care. Six electronic databases were searched for relevant studies up to August 2022. A narrative synthesis of findings was conducted to evaluate the intervention processes and results. Two independent reviewers assessed the relevance, risk of bias and quality of the evidence. A total of ten studies were included (eight RCTs and two non-RCTs). Overall, seven studies showed a slight reduction in AP. However, this reduction in AP when offering a dashboard may not in itself result in reductions but only when combined with educational components, public commitment or behavioural strategies. Only one study recorded dashboard engagement and showed a difference of 10% (95% CI 5% to 15%) between intervention and control. None of the studies reported on the development, pilot or implementation of dashboards or the involvement of stakeholders in design and testing. Interactive dashboards may reduce AP in primary care but most likely only when combined with other educational or behavioural intervention strategies.

## 1. Introduction

The rise in antibiotic consumption has resulted in the spread of antimicrobial resistance, which was responsible for 1.27 million deaths in 2019 [1,2]. Up to 20% of antibiotic prescribing (AP) is deemed to be inappropriate, translating to 20,000 unnecessary APs in the UK daily [3,4]. Despite extensive efforts to promote prudent use of antibiotics, ambulatory AP has only slightly decreased over the past decade and this positive trend varied between countries [5]. In Europe in 2020, the AP rate was 600 per 1000 persons per year, while in the US, this was around 800 prescriptions per 1000 persons per year [5]. In the EU/EEA, an overall reduction from 19.3 (2012) to 15 (2020) daily doses (DDD) per 1000 inhabitants per day was recorded, which translates to a 22% reduction in antibiotic consumption in the community. Bulgaria was the only country where total antibiotic consumption over this period increased [6].

The introduction of technology to optimise prescribing and quality improvement coincided with the introduction of clinical decision support systems (CDSSs) and audits and feedback (A&F) [7]. CDSSs provide information on best evidence guidelines to close the gap between optimal practice and actual clinical care [8]. However, CDSSs have shown to have low to moderate effects on improving appropriate AP [9]. CDSS implementation in primary care has shown workflow barriers resulting in alert fatigue and negative experiences (due to limit the prescriber to approved treatment option) by the General Practitioner (GP) [10,11,12].

On the other hand, A&F systems do not interfere with doctor prescribing autonomy and deliver options for education through a feedback tool [13]. Traditional A&F interventions utilising peer comparison, where individuals are compared to top-performing peers, along with positive reinforcement, have shown a decrease of 16% of AP in primary care [14]. Additionally, A&F interventions often provide a visual element or dashboard to deliver clinical performance feedback [14,15]. In 2015, Dowding et al. published a comprehensive overview of clinical and quality dashboards in healthcare environments for general treatments and concluded that introducing dashboards can positively decrease ventilator-associated pneumonia rates, increase on time AP and improve turnaround time for signing reports. However, the relationship between dashboard features (graphical display type and presentation methods to users) and improvements in outcomes or incorporation into everyday clinical practice was unclear [16]. A more recent systematic review (SR) of randomised controlled trials (RCTs) that evaluated the use of clinical dashboards integrated in patient management systems showed improved medication adherence (for patients with inflammatory arthritis) and test ordering (cardiovascular risk screening of patients). However, this review also reported limited impact of dashboards on the prescription of antibiotics and statins [17].

With increased digitalisation of healthcare, new technologies and improved data visualisation techniques, advances have been made to integrate A&F and CDSSs into clinical care [18,19]. Today, some patient management systems include interactive dashboards with or without integrated A&F or CDSS. This review aims to assess the effectiveness of interactive dashboards to optimise AP in primary care.

## 2. Results

After removing duplicates, a total of 6539 potentially relevant reports were recovered. After evaluating 47 full text reports, 10 studies [20,21,22,23,24,25,26,27,28,29] were included in the synthesis of evidence (Figure 1). The list of excluded studies and the reasons for their exclusion are shown in Appendix A. The characteristics of the included studies are detailed in Appendix A.

### 2.1. Included Studies

#### 2.1.1. Study Design

Ten studies were included: three individual RCTs [20,26,28], four cluster trials [22,23,25,27], a crossover trial [24]) and two non-RCTs (one a controlled before and after study [21] and one ITS [29]) (Appendix A).

#### 2.1.2. Participants and Settings

The number of participants was reported in seven studies (individual level). The participants included 3609 physicians [20,21,24,26,27,28] and 2566 dentists [25]. The study setting included general practices [22,23], dental practices [25], primary care institutions [24,27,29], community health centres [23], community-based practices [23], hospital-based practices [23] and emergency departments (ED) [21] (Appendix A). The ED was included as a primary care setting as some countries provide GP/primary care services within the ED [31].

#### 2.1.3. Description of the Intervention

The characteristics of the included intervention and control studies are available in Appendix A. Overall, ten studies used a visualisation tool or dashboard to provide A&F of AP to the prescriber, and the duration of the interventions varied between three months to two years [20,21,22,23,24,25,26,27,28,29]. However, six of these added other elements to the intervention. Dun Yan (2021) included an education component, which provided the national consensus treatment guidelines and an online education course [26]. Hemkens (2017) included recommendations for evidence-based guidelines for optimised antibiotic use in primary care [28]. Curtis (2021) included three different waves in the behavioural impact intervention for optimised engagement: a tailored broad-spectrum antibiotic feedback to which in wave 2 a reminder (dashboard link) was added and in wave 3 a potential cost saving [22]. Shen (2018) added information on operation guidelines, public commitment and takeaway information (patients take home) [27]. Elouafkaoui (2016) provided a behaviour-change message, which was created with guidance recommendations for AP [25]. Davidson (2022) included an education campaign for patients and providers [29]. Across the 10 studies, the control group consisted of education components only [26], usual static email attachments [20,27] or no intervention (usual care) [20,21,22,23,24,25,28].

Table 1 describes the visualisation tools or dashboards. Daneman (2021) [20], Linder (2010) [23] and Davidson 2022 [29] briefly mentioned details of dashboard development; however, this process varied widely between these studies. The data for the dashboard and main outcome measurements were predominantly from patient management systems. However, Hemkens (2017) used data from statutory health insurers for drug prescription and health care service claims [28] and Curtis (2021) used data from national monthly datasets [22]. The data summaries and features varied among studies, describing AP by diagnosis [21,23,24,26,27,29] or type of antibiotic [23,24,28,29]. Furthermore, some included peer comparison [20,21,24,25,27,28], information of other medications [20] and practice overviews [20,23,26,28,29]. Five studies included reminders about using the dashboard at ten days and up to six months [21,22,23,24,28].

#### 2.1.4. Outcomes

Eight studies measured changes in AP (primary outcome) but used different outcomes and measure types (Appendix A). Four studies reported changes as overall rate of AP [24] by diagnostic categories (acute respiratory infections [21,23], upper respiratory infection, bronchitis, sinusitis and pharyngitis [26]). Hemkens (2017) reported change from baseline (between-group difference) in AP per year (defined daily doses-DDD/100c) according to patient group, type of antibiotic and age group [28]. Elouafkaoui (2016) reported the change from baseline of all antibiotic items/100 claims and defined daily dose (all antibiotics)/100 claims [25]. Shen (2018) reported the percentage of patients that received an antibiotic when presenting with symptomatic respiratory tract infections or gastrointestinal tract infections [27]. Curtis (2021) measured the proportion of broad-spectrum antibiotics out of the total number of antibiotics prescribed [22].

Regarding secondary outcomes, Linder (2010) and Jones (2021) reported appropriate and inappropriate antibiotic use outcomes [21,23] and Davidson (2022) reported inappropriate antibiotic use [29]. These outcomes were defined by Linder (2010) as antibiotic appropriate and inappropriate Acute Respiratory Infection (ARI) visits when compared to the International Classification of Diseases (ICD-9-CM) code for pneumonia, streptococcal pharyngitis, sinusitis and otitis media [35]. Jones (2021) followed the Meeker et al. [36] methods to develop a list of ICD-10 Codes for upper respiratory system conditions for which antibiotics were considered appropriate and inappropriate (clinician experts reviewed the list) and also mentioned using evidence-based guidelines for when antibiotics were appropriate (however, they did not reference which guidelines were used) [21]. Davidson (2022) included conditions (ICD-10) for which antibiotics are not indicated (acute sinusitis, otitis media (nonsuppurative), acute bronchitis, pharyngitis (nonbacterial), cough, upper respiratory infection (URI), common cold, allergic rhinitis and influenza) [29].

Du Yan (2021) described the proportion of total visits where an antibiotic was prescribed for sinusitis or pharyngitis over time [26]. Daneman (2021) included prolonged antibiotic duration (proportion of antibiotic treatments exceeding 7 days during the quarter (four quarters of 2018 and four quarters of 2019)) and antibiotic initiation, defined as the proportion of residents initiated on an antibiotic during the quarter [20]. Two studies reported dashboard engagement, Curtis (2021) recorded practices having at least one dashboard view [22] and Linder (2010) measured the proportion of intervention physicians who used the dashboard at least once [23] (Appendix A).

### 2.2. Excluded Studies

Of the 53 possible eligible studies, 40 were excluded after assessing the full text. See Appendix A of the characteristics of excluded studies in full text and the PRISMA study flowchart Figure 1.

### 2.3. Effects of the Interventions

The summary of the results is described in Appendix A. There was high heterogeneity due to varying definitions of the outcome, measure types, type of infection and the combination of other interventions. As a result, an overall pooled effect outcome could not be determined. Figure 2 shows the effect of AP changes (primary outcome) in odds ratios, for three studies reporting this outcome. The results illustrate a larger decrease in AP in the intervention group compared to the control group, except Du Yan (2021) [26] for sinusitis and pharyngitis and Linder (2010) who compared the ARI Quality Dashboard versus usual care [23]. However, in Linder (2010), the intervention physicians who used the dashboard at least once were less likely to prescribe antibiotics for all ARIs compared to those who did not use the dashboard [23]. It is notable that only Linder (2010) sent monthly e-mails reminding physicians of the dashboard (Table 1) [23].

Figure 3 illustrates the effect of AP changes quantified as the percentage change from baseline (between-group difference) for two studies and no overall differences in AP were observed. However, Hemkens (2017) showed that quarterly written personalised prescription feedback was associated with reduced AP in the 6 to 18 year age group in the first year (−9% (CI 95% −15% to −2%)) and in adults (19 to 65 years) in the second year (−5% (CI 95% −8% to −1%)) [28]. Elouafkaoui (2016) reported a 6% reduction in AP in the intervention group (written behaviour change message with peer comparison and A&F relative) to the control group (no A&F). Appendix A describes the result of other intervention subgroups of this factorial study. Elouafkaoui (2016) was the only cluster study that reported an intra-cluster correlation coefficient (0.2 (CI 95% 0.1 to 0.2)). Furthermore, the interval between receiving A&F varied according to allocation (either 0 and 6 months or at 0, 6 and 9 months) [25].

Curtis (2021) found that the addition of reminders and information on impact of prescribing to tailored broad-spectrum antibiotic feedback (behavioural impact intervention) resulted in lower broad-spectrum prescribing by comparison with feedback without reminders (plain intervention) (plain versus behavioural intervention, regression coefficient 0.0041, CI 95% 0.00007 to 0.008). However, when both interventions were compared to no intervention, no difference in AP reduction was shown (Appendix A). The cross-over study (Chang 2020) reported no significant difference in AP between the intervention comprising feedback and individual ranking score, with no intervention, which may be due to a carryover effect (see RoB section). The controlled before and after study (Jones 2021) showed a reduction in AP of 5% for the intervention group (from 36% at baseline to 31% during follow-up), compared with an increase of 3% in the control group (from 38% to 41%) (Appendix A) [21].

Appropriate and inappropriate prescribing was assessed as a secondary outcome (see definitions in outcome section) in Linder (2010), but no differences were observed between ARI Quality Dashboard and the control [23]. Jones (2021) showed less inappropriate AP in the intervention with 22% at baseline and 15% during follow-up compared to 23% and 24% in the control [21]. Davidson (2022) reported a significant reduction of 19% in inappropriate AP before and during the intervention period [29].

Daneman (2021) reported no difference in the antibiotic duration and antibiotic initiation in the intervention group (dashboard tool) compared to the control group (usual static email attachments) [20]. For dashboard engagement outcomes, Curtis (2021) observed a difference of 10% (95% CI 5% to 15%) between the intervention and control group in the number of times they interacted with the dashboard [22]. The effect of the intervention on other secondary outcomes is described in Appendix A.

### 2.4. Risk of Bias

The RoB is summarised in Figure 4 for the eight RCTs, while the controlled before and after study and ITS are described in Appendix A. RoB of the RCTs from Daneman (2021) and Hemkens (2017) had a low overall RoB [20,28]. The Du Yan (2021) study showed bias due to deviations from the intended intervention, measurement of outcome and selection of the reported result, because no participants were blinded, outcome assessors were aware of the intervention received by study participants, and the trial protocol was not available [26].

For the cluster RCTs, Shen (2018) had a high RoB due to the sequence generation process, the allocation sequence concealed, measurement of the outcome and selection of the reported result [27]. Two cluster RCTs had deviations from the intended interventions (effect of assignment to intervention) [22,25]. Linder (2010) reported challenges with the randomisation process and selection of the reported result, and the trial protocol was unavailable [23].

The crossover RCT from Chang (2020) had a high RoB arising from period and carryover effects (no washout period) [24]. Finally, the two non-randomised studies (Jones (2021) a controlled before and after study [21] and Davidson (2022) an ITS [29]), showed moderate RoB due to confounding and lack of protocol to evaluate or pre-specify the methods (Appendix A).

### 2.5. Grading the Quality of Evidence

Based on the assessment of eight studies (Appendix A), the overall level of certainty in the evidence (GRADE) for the primary outcome (overall change of AP) was judged at low (Appendix A).

## 3. Discussion

This SR identified 10 studies that evaluated the impact of dashboards or visual analytical tools to optimise AP. Overall, seven studies indicated a slight reduction in AP [21,22,25,26,27,28,29] and six of these added other elements to the intervention [22,25,26,27,28,29]. Interventions that included an educational component (including national guidelines), public commitment, behavioural strategies (reminder, cost saving, written behaviour change message) were more likely to show an impact. Appropriate AP showed improvements between the intervention and control groups or over follow-up periods. The two studies that assessed dashboard engagement, one showed improved AP with increased engagement.

Our findings are consistent with a previous SR of Xie et al. (2022), which assessed the use of clinical dashboards on medication prescription and testing ordering [17]. Xie et al. reported limited evidence of dashboards in relation to medication adherence in general, reduction of opioids, AP and test ordering [17]. Furthermore, dashboard interventions are frequently part of multifaceted interventions to improve and generate positive changes in healthcare outcomes, making evaluating of each component challenging to separate [17]. Our findings complement the SR of Xie et al. extending the number of studies focused on AP (three RCTs [20,22,27] and two non-RCTs [21,29]), more detail description of the visualisation tool or dashboard and restriction to one priority setting (AP and primary care). Nevertheless, the level of certainty of evidence assessment differs from the SR of Xie et al. [17] which judged the primary outcome individually between moderate and very low quality [17] using the standard-GRADE approach [37] without considering inconsistency and publication bias. For our primary outcome (overall changes in AP) certainty was low using a GRADE approach focused on SRs summarised narratively without meta-analysis [38].

To implement an interactive dashboard, good knowledge of operating systems and user interaction, context factors, barriers and facilitators is required, combined with effective use of data, the application of best knowledge and continuous improvement [39]. New methods have a learning cycle of evaluation and monitoring users’ technological competence [11] which cover interaction effectiveness, user experiences and system efficacy [40]. The studies included in this SR do not describe the development and design of the dashboard and only include a short description (Table 1). In addition, only two studies reported the inclusion of qualitative and mixed method perspectives in the design and implementation phase. Daneman (2021) [20] explored how A&F influence AP through semi-structured interviews with prescribers [32]. Shen (2018) [27] used preliminary results of mixed methods [41] and qualitative research [42] to develop and adapt the intervention.

In any behavioural intervention, the situational context of delivering information is essential [36,43] and for feedback to be effective, it should be frequent, individualised and available at the recipients request [43]. An interesting aspect highlighted by Tasang et al. (2022) in their SR of computerised A&F systems in healthcare is to allow the feedback to be actionable (specific to user roles). However, organisational context, resources and user characteristics influence potential effects of A&F systems considerably [14]. Therefore, it is relevant to consider involving various perspective levels, including organisational, patient and public [44] in these systems, as Shen (2018) and Davidson (2022) briefly incorporated. Shen (2018) integrated a component of the intervention targeting patients and the general group through public commitment and detailed patient information explaining diagnosis, treatment, symptom relief and activities to prevent future infection [27]. Davidson (2022) included an antimicrobial stewardship education campaign for patients and providers which included dashboards. Moreover, both had access to information from consumer web pages and multimedia pitches on Facebook, Twitter, Instagram and other social media platforms [29]. In general, without regular individual engagement in antimicrobial stewardship strategies, there are no shortcuts to improving AP and using digital tools only seems to be insufficient.

A limitation of this SR was the high heterogeneity of the included studies, which did not allow an overall conclusion through meta-analysis to be determined. Nonetheless, the findings were consolidated, aggregated, and narratively reported. Additionally, an acceptable technique was employed to assess the certainty in the evidence of our primary outcome to make informed judgements. Another constraint was publication bias, which was mitigated by scanning electronic databases comprehensively and using other approaches, but it still remains a constraint.

## 4. Materials and Methods

The protocol of this systematic review was registered in PROSPERO (CRD42022313006) and followed the Preferred Reporting Items for Systematic reviews and Meta-Analyses (PRISMA) guideline [30].

### 4.1. Criteria for Considering Studies for this Review

Participants: general practices and primary care settings focused on GPs or other health professionals.Intervention: Any intervention using prescription data illustrated in a visual analytical tool (i.e., dashboard). Decision support tools which were incorporated as alerts or risk calculators were excluded.Comparator: usual care or any other intervention without visual analytical tools (dashboard).Outcomes of interest included:○Change in AP (primary outcome)○Prescribed antibiotic class○Change in prescription of inappropriate (i.e., not recommended) antibiotics○Antibiotic duration○Patients’ re-consultation○Dashboard engagement (not initially included in the protocol).Types of studies: RCTs and non-randomised controlled trials (non-RCTs) (controlled before and after studies, interrupted time series studies (ITS) and controlled trials using non-random methods) assessing the effectiveness of dashboards including prescription data in general practice.

### 4.2. Search Methods for Identification of Studies

A systematic search strategy was carried out in the following electronic databases: Cochrane Central Register of Controlled Trials (OVID), MEDLINE (OVID), EMBASE, SCOPUS, Web of Science Core Collection, and LILACS (Appendix A). Reference lists of included studies, google scholar and relevant webpages were searched for additional papers. The cut-off point for inclusion was 15 August 2022.

### 4.3. Data Collection and Analysis

#### 4.3.1. Selection of Studies

Researchers (N.G.-O., S.P., D.A. and H.V.) independently (double) screened the abstracts and titles of articles retrieved from each search to assess eligibility. Copies of the full text of all eligible papers were obtained and independently evaluated by two researchers (N.G.-O., S.P., D.A. and H.V.) according to the prespecified selection criteria. In case of discrepancies, a third researcher (A.V.) provided resolution. Authors of abstract of conferences and protocols were contacted to ask for results and full text.

#### 4.3.2. Data Extraction and Management

The data extraction form included the study ID, available information, study eligibility, a summary of assessment for inclusion, population and setting, methods, the risk of bias assessment, intervention and control groups, outcomes, results, limitations, conclusions of study authors and funding. One researcher (N.G.-O.) extracted the data which was checked by a second researcher (S.P., D.A. and H.V.).

#### 4.3.3. Risk of Bias (Quality) Assessment

The risk of bias (RoB) of the RCTs (individual, cluster and crossover trial) was assessed independently by two researchers (N.G.-O., S.P., D.A. and H.V.) using the Cochrane “RoB 2” tool [45,46,47], with non-RCTs assessed with the ROBINS-I tool [48]. This assessment was based on the effect of the assignment to intervention (the ‘intention-to-treat’ effect) and focused on the primary outcome of each study. Disagreements were resolved by discussion.

#### 4.3.4. Measures of Treatment Effect

Dichotomous outcomes were expressed as odds ratio (OR) or absolute risk difference (AR) with 95% confidence intervals (CI). Continuous data were reported as changes from baseline through mean differences (MD) with 95% CIs. Meta-analysis was considered but could not be performed due to the high heterogeneity between studies. When the study only reported the outcome in a proportion or percentage, an OR was estimated (fixed effect) in Review Manager 5.4.1 [49].

#### 4.3.5. Missing Data

The data extraction form captured information on missing outcome data from each study. Imputation methods, where applied, were recorded.

#### 4.3.6. Assessment of Heterogeneity

The evaluation of heterogeneity (I2 statistic) was not possible due to the small number of included studies with the same type of study and outcome. Results are illustrated in a forest plot, excluding an overall pooled effect diamond, using R software. Heterogeneity was assessed through a visual and qualitative assessment.

#### 4.3.7. Data Synthesis

Measure effect sizes were presented for each study (with a range). A narrative synthesis of findings was conducted to assess the intervention processes and results.

#### 4.3.8. Grading the Quality of Evidence

The quality of evidence for our primary outcome was assessed using the constructs of the GRADE “Grades of Recommendation, Assessment, Development, and Evaluation” approach for SRs summarised in a narrative without meta-analysis, which include RoB, imprecision, indirectness, inconsistency and risk of publication bias [38].

## 5. Conclusions

Dashboards visualising health data may reduce AP in primary care, but this is generally only in combination with frequent and individualised feedback. None of the included studies reported on the dashboards’ development and implementation phase and it was unclear whether users were involved in delivering tailored dashboards. Future research in the use of interactive dashboards to optimise AP in primary care should consider and report on operating systems, user interaction, involvement of stakeholders in design and testing, context factors, barriers and facilitators for implementation and sustainability of these visualisation tools.

## Figures and Tables

**Figure 1 antibiotics-12-00136-f001:**
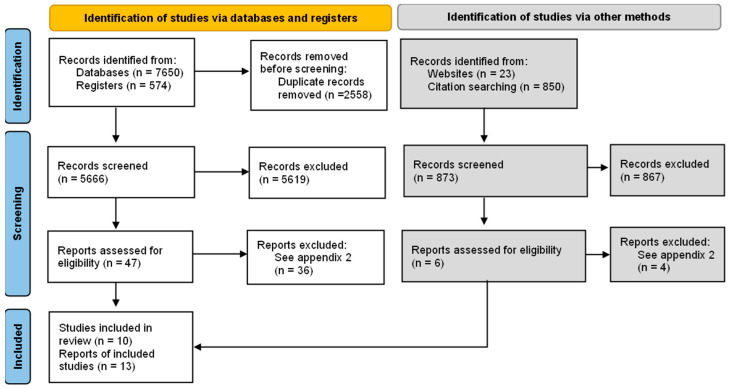
Flowchart of studies selection. Source: The PRISMA 2020 statement: an updated guideline for reporting systematic reviews [30].

**Figure 2 antibiotics-12-00136-f002:**
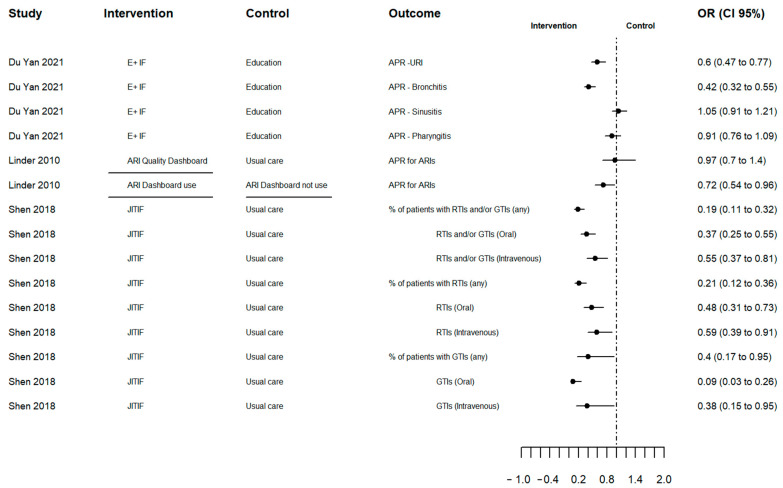
Forest plot of antibiotic prescribing change outcome in odd ratios. Source: elaborated by the authors with information reported of studies included. E+F: Education plus individualized prescribing feedback dashboard; ARI: Acute Respiratory Infection; ARI Quality Dashboard Use (line segment): intervention physicians who used the ARI Quality Dashboard at least once versus intervention clinicians who did not use it; JITIF: Just-in-Time Information and Feedback; APR: Antibiotic prescription rate; URI: Upper respiratory infection; RTIs: Respiratory tract infections; GTIs: Gastrointestinal tract infections; OR: Odd ratio; CI: confidence interval.

**Figure 3 antibiotics-12-00136-f003:**
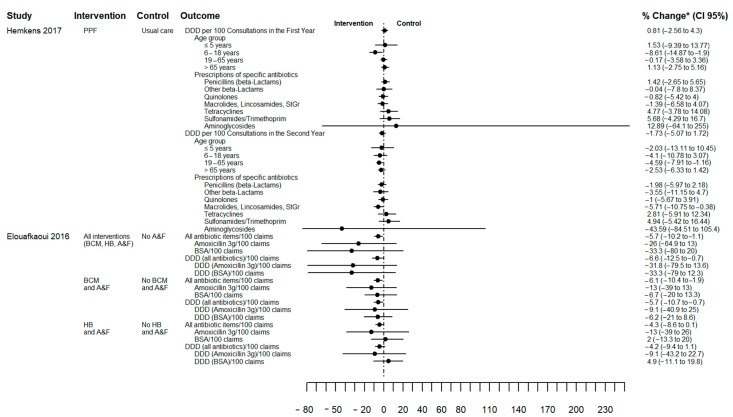
Forest plot of antibiotic prescribing change outcome in percentage of change from baseline (between-group difference). Source: elaborated by the authors with information reported of studies included. PPF: Personalised prescription feedback; BCM: written behaviour change message; HB: health board; A&F: audit and feedback; BSA: Broad spectrum antibiotics (clindamycin, co-amoxiclav, clarithromycin, cefalexin, and cefradine); DDD: defined daily doses; CI: confidence interval. * Hemkens (2017) reported change from Baseline (between-group difference) [28] and Elouafkaoui (2016) reported change from baseline with all percentages standardised using control group baseline mean prescribing rate [25] (see Appendix A).

**Figure 4 antibiotics-12-00136-f004:**
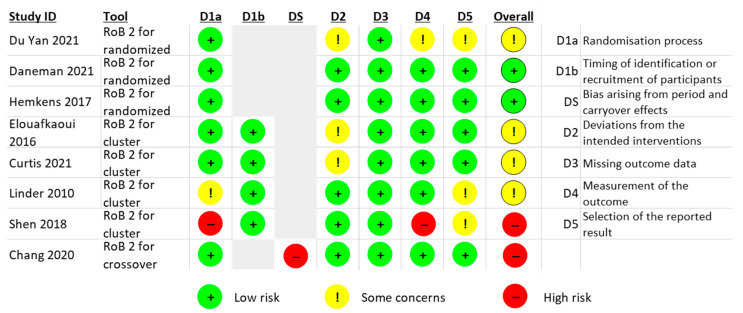
Risk of bias (RoB) summary. Source: elaborated by the authors.

**Table 1 antibiotics-12-00136-t001:** Description of dashboard or visualisation tools included of interventions groups of studies included.

Study ID	Data Summarized	Features	Development Details	Extracted Data from	Time Period of Report	Access	Engagement and Reminder Strategies
Du Yan 2021 [26]	Rate of antibiotic prescription (AP) and practice-wide prescribing rates for upper respiratory infection (URI), bronchitis, sinusitis, and pharyngitis.	Personalised for each clinician, including a practice summary (practice’s antibiotic prescription rates for target conditions), individual clinician prescription and the difference with their practice.	No detail	Electronic medical record without a separate database.	Report from previous month starting May 2018.	An online dashboard; the paper provided a sample in a figure (see Figure 2 from original paper [26]).	No detail
Daneman 2021 [20,32]	Percentages of AP and prolonged antibiotic treatment (longer than seven days). Additionally, antipsychotic, benzodiazepine, and other neurotropic medication prescribing was reported.	A home page (overview) with key messages from prescribing data, peer comparisons (question mark icon if prescriptions were higher, similar or lower than their peers), and two links (to view trend data and change ideas). The antibiotic page allowed comparing their overall rate with Ontario percentiles, showed key changes and answers to important questions (relating to resident characteristics, accurate data, the rate calculated, data limitation, and low AP that was reasonable and safe).	Input from infectious diseases, implementation science, information technology, and quality improvement specialists to improve its design through an iterative, user-centered design process.	Administrative health databases and linked with drug, hospitalization, and emergency department databases.	Four quarters of 2018 and four quarters of 2019	An online dashboard; the paper provided a screenshot of a sample in a supplement (see Supplement S1 from original paper [20])	No detail. However, they explored how the intervention was perceived by those that engaged with it in its qualitative study [32].
Hemkens 2017 [28,33]	Antibiotic prescriptions per 100 consultations in the preceding months and displaying the adjusted average in peer physicians, that is, the entire population of Swiss primary care physicians.	Details on the prescriptions per age group or sex or for certain antibiotic types and answers to frequently asked questions on antibiotic use.	No detail	Data from statutory health insurers for claims of drug prescriptions and health care services.	Quarterly intervals (not more report details)	An online dashboard; the paper provided a screenshot of a sample in a supplement (see Supplement Figure from original paper [28])	Physicians received a quarterly updated personalised prescription feedback
Curtis 2021 [22]	Change in AP	No detail	No detail	National datasets published monthly by NHS Digital (Practice-level prescribing data).	No detail	An online dashboard with a single measure highlighted (a link to their practice dashboard on OpenPrescribing.net.). The study provided a sample image in Supplement (see supplement Figure S1 from original paper [22])	No detail. However, an update was sent at 5-week intervals.
Linder 2010 [23]	The proportion of Acute Respiratory Infection (ARI) visits with antibiotics, the proportion of individual ARI diagnoses (pneumonia, sinusitis, acute bronchitis) with antibiotics, the proportion of broad-spectrum AP, the distribution of ARI visits by evaluation and management billing codes, and individual patient visit details.	Design based on the recommendations of the Centers for Disease Control and Prevention and the American College of Physicians. ASP.NET technology used to build the Dashboard. Option to “drill down” to any patient’s medical record directly from the Dashboard to review patient details and export the report for additional follow-up or analysis.	A pilot to assess the users access, understand if it was useful to their antimicrobial prescribing patterns and validate its reports with primary data from the HER by drilling down to individual patient charts.	Electronic health records (EHR)	Dashboard displayed visit and prescribing data for the previous year and was automatically updated monthly.	Physicians accessed the Dashboard from the EHR Reports Central area, which contained about 10 other reports about preventive and chronic disease management. The study provided a screenshot of the dashboard in Figure (see Figure 1 from original paper [23])	Monthly e-mails reminding physicians about the ARI Quality Dashboard.
Shen 2018 [27]	Their performance scores (PSs) and percentages of prescribed antibiotics use (ABU).	The PS and ABU were presented in red, yellow, and green, respectively, if it fell below (or above), within, and above (or below) the interquartile range of the same PS or ABU). Additionally, it illustrated relevant performance feedback, performance scores for current doctor and their peers in total and by infections, public commitment, bulleted points of commitment letter, and frequently questions.	No detail	Data was based on the records of their management of symptomatic infection patients	No detail	Web-based aid (WBA). A slide of WBA in Multimedia Appendix (see appendix A3 from original paper [27])	No detail
Elouafkaoui 2016 [25,34]	Prescribing rate number of antibiotic items dispensed multiplied by 100 claims and the health board rate (the overall ordinary list prescribing rate for current dentists in non-salaried practices in NHS Example Board)	No detail	No detail	Electronic healthcare datasets held centrally by the Information Services Division of NHS National Services Scotland.	Monthly	This Audit and Feedback included a visualisation (line graph) which was delivered by post. The study provided an example in Supplement (see supplement Figure S1 from original paper [25])	No detail
Chang 2020 [24]	An individual ranking score of AP (peer comparison), statistic information about the diagnosis and AP (total and type of antibiotics).	Top of the screen: the top five diseases of patients seen by the physician over the previous 10 days, the start and stop time for the previous 10 days, and the number of prescriptions given during this period and department ranking Bottom: Statistics on the antibiotic frequency, prescription rate of each antibiotic prescribed, precautions and contraindications for antibiotics being used.	No detail	Health information system (HIS).	Previous 10 days	A link on HIS to see the feedback information any time. The paper provided an example of feedback information displayed on a physician’s computer screen in Figure (see Figure 1 from original paper [24])	A pop-up window to automatically prompt to check for the feedback information every 10-days
Jones 2021 [21]	Rate of inappropriate prescribing and stratified by diagnosis category.	Top peer comparison (top 10% of performers (clinicians with the lowest prescribing rates) or not to be among the 10% best performers). Rolling over each column shows the percentage for each provider and the number of encounters on which the rate of inappropriate prescribing is based. Filters allow the provider to compare data over different timelines and across departments.	No detail	Electronic health record system.	Unclear, but it mentioned “Dashboard was updated daily”	Tableau dashboard. The paper provided a figure of provider feedback dashboard in Supplementary (see supplementary Figure S2 from original paper [21])	Physicians’ review of their personal data was structured to satisfy the requirements for the American Board of Emergency Medicine Maintenance of Certification Improvement in Medical Practice Requirements. Physicians received biannual e-mails
Davidson 2022 [29]	Prescribing rate, target rate, antimicrobial encounters, total encounters and antimicrobial prescribing rate	Comparing AP behaviours among providers, practices and organisational groupings. Data viewable by indication, antibiotics class, and at the levels of provider, practice site, specialty medical director and administrator.	Developed in Microsoft Power BI. Including coding, targeted indicators, instructional webinar and on-site dashboard navigation education given upon request for practice sites and leaders. The dashboard remains part of continuous, ongoing assessment of feedback from users and leadership.	Electronic health record and administrative data sources.	Prescribing data compared year-to-year and rolling 12 months	Online dashboard; the paper provided a figure of Dashboard Overview in Supplementary (see supplementary Figure S8 from original paper [29])	Antibiotic education campaign (provider focused resources)

Source: elaborated with information reported of studies included.

## Data Availability

Not applicable.

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
