# Peer review of "The Effectiveness of Interactive Dashboards to Optimise Antibiotic Prescribing in Primary Care: A Systematic Review"

_antibiotics, 2023, doi:10.3390/antibiotics12010136_

Round 1
Reviewer 1 Report
Dear authors,
Thank you for very interesting review of the effectiveness of interactive dashboards in optimisations of AP in primary care settings.
This review of very heterogeneous interventions has shown the difficulties in AP improvement and the need to multifaceted interventions. To my opinion the important massage of this paper is the fact that without regular individual engagement of the AMS expects also in the primary care settings there are no shortcuts in improving AP and a usage of digital tools only is not sufficient.
Best regards
Author Response
We thank the reviewer and agree with their conclusions. We added a sentence to the discussion section to emphasise their comment, page 13, lines 375 to 377, which now reads:
“In general, without regular individual engagement in antimicrobial stewardship strategies, there are no shortcuts to improving AP and using digital tools only seems to be insufficient”

Reviewer 2 Report
Useful study demonstrating important results.
Very minor changes needed:
Line 80. "Comparator: usual care or other any other...."- re-wrte
Line 95. "A systematic search strategy was carried out on in the following...." - re-write
Line 155. "Ten studies were included of which three individual RCTs..."
Change to:Ten studies were included: three individual RCTs...
Author Response
We thank the reviewer for these suggestions and changed accordingly:
Line 80: “usual care or any other intervention”
Line 95: “A systematic search strategy was carried out in the following....”
Line 155: “Ten studies were included: three individual RCTs”

Reviewer 3 Report
Thank you for the opportunity to review this manuscript.
I see this manuscript is very well written. It is only possible in the table section to be rearranged so that the table alone can be read. And only things that should be highlighted are shown in the form of a description
Author Response
We thank the reviewer for the approval.
It is not clear to us if the reviewer is requesting a change to the table, and if so, how this should be done.
